# Taxonomy Complexity of Some Tyrrhenian Endemic *Limonium* Species Belonging to *L. multiforme* Group (Plumbaginaceae): New Insights from Molecular and Morphometric Analyses

**DOI:** 10.3390/plants11223163

**Published:** 2022-11-18

**Authors:** Duilio Iamonico, Olga De Castro, Emanuela Di Iorio, Gianluca Nicolella, Mauro Iberite

**Affiliations:** 1Department of Environmental Biology, University of Rome Sapienza, P.le A. Moro 5, 00185 Rome, Italy; 2Department of Biology, University of Naples Federico II, Botanical Garden, via Foria 223, 80139 Naples, Italy; 3DNATech Srl, Spin-Off Company of the University of Naples Federico II, Botanical Garden, via Foria 223, 80139 Naples, Italy

**Keywords:** morphometry, new subspecies, Latium, taxonomy, nuclear DNA, plastid DNA, phylogeny, phylogeography

## Abstract

The delimitation of *Limonium* taxa is highly complicated due to hybridization, polyploidy, and apomixis. Many “microspecies” were described and aggregated into groups, most of which are still poorly known from both molecular and morphological points of view. The aim of this study is to investigate four endemic species from the Tyrrhenian coast of central Italy and the Ponziane Archipelago belonging to the *L. multiforme* group (*L. amynclaeum*, *L. circaei*, *L. pandatariae*, and *L. pontium*) by means of molecular and morphometric analyses. Molecular data by sequencing ITS and three plastid markers and morphometric data highlight new information about the taxonomy of these taxa so as to reduce them into a single specific entity. In fact, the better taxonomic choice is to consider the populations studied as part of a single species, i.e., *Limonium pontium*. Three subspecies are recognized, i.e., subsp. *pontium* [= *L. circaei* = *L. amynclaeum*; from Circeo to Gianola localities (excluding Terracina) and from islands Ponza, Palmarola, Zannone, and Santo Stefano], subsp. *pandatariae comb. et stat. nov.* (from island of Ventotene), and subsp. *terracinense subsp. nov.* (from Terracina).

## 1. Introduction

*Limonium* Mill. is the largest genus of the family Plumbaginaceae Juss., comprising more than 600 species which naturally occur in all the continents except Antarctica [1]. The Mediterranean Basin is one of the centers of diversity of this genus and most of the currently accepted species are concentrated there [2]. Members of *Limonium* are important constituents of halophytic communities growing in coastal areas in rocky and sandy places as well as in salt marshes (e.g., [2,3,4]), having a significant role in the biodiversity of these places [5].

The delimitation of species within the genus *Limonium*, which is crucial for their conservation, is, however, complicated due to several phenomena, i.e., hybridization, polyploidy, and apomixis, at least in some species groups [4,5,6,7,8,9,10]. According to some authors (e.g., [2]), these evolutionary processes would occur in the typical habitat in which *Limonium* grow (especially in insular and peninsular areas) because these places are often geographically and/or ecologically isolated, leading the segregation of many microspecies. In addition, a particular reproductive strategy occurs in *Limonium*, as first noted by Baker [11] who demonstrated that the heterostyly pollen/stigma dimorphism is linked to the occurrence of apomixis, that is: the sexual species are characterized by having a dimorphic self-incompatibility system, whereas the agamospermous species are almost always monomorphic. Finally, as a consequence of these segregation events, the intricate nomenclature of the genus causes further complications in the understanding its taxonomy and the species delimitation [12,13,14,15,16,17,18].

Concerning karyology, *Limonium* shows high levels of polymorphisms linked with both polyploidy (triploids (mostly occurring in the western Mediterranean area) to octoploids (tetraploid to octoploids mainly occur along the Atlantic coasts and in the eastern Mediterranean region)) and aneuploidy [5,19,20,21,22].

On the basis of the available molecular data (e.g., [4,23]), two subgenera are currently recognized, i.e., *Limonium* subgen. *Limonium* and subgen. *Pteroclados* (Boiss.) Pignatti. At sectional and species levels, the situation is more difficult, especially for the species-rich Mediterranean group, as highlighted by Malekmohammadi et al. [4] who stated that additional studies are required. In particular, the concentration of a large number of morphologically differentiated taxa has resulted in the description of myriad microspecies (predominantly agamospecies and often sympatric) complicating the systematics of the genus [2].

The flora of Italy currently comprises 111 *Limonium* taxa of which 99 are endemics [24]. All in all, Italian *Limonium* species represent 24% of the total number of species (469) occurring in the Euro + Med area [25]. Many of these species are considered as “microspecies”, i.e., mostly apomictic taxa [10] which are distinguished from each other by a few morphological differences and often have a restricted distribution area (sometimes even reduced to locus classicus only). As a consequence, aggregates were created to group similar microspecies (see [26]). Anyway, many of these groups are still poorly known, especially in the molecular context (see, e.g., [2,26]). Among the most complicated *Limonium* aggregates, the “*L. multiforme* group” includes 16 species occurring along the Tyrrhenian coast of central Italy and the associated islands [27]. The flora of the Lazio region comprises four endemic species which belong to the *L. multiforme* group (*L. amynclaeum* Pignatti, *L. circaei* Pignatti, *L. pandatariae* Pignatti, and *L. pontium* Pignatti; [28,29]) and, based on the available literature, they are very similar from the morphological point of view and, as a consequence, their distribution is still not well known at present. No karyological counts are available for these four endemic species. According to Brullo and Guarino [27], since this group is represented by diploid species with 2n = 18 (with the one exception of *L. hermaeum* with 2n = 27 [30]), the populations are probably interfertile and only the geographical isolation has led to the current differentiation, favoring the processes of speciation. So, as part of the ongoing studies on the genus *Limonium* [13,14,15,16,17,18,31], we here present a study aiming to clarify the taxonomic value of these four endemic Tyrrhenian species (i.e., *L. amynclaeum*, *L. circaei*, *L. pandatariae*, and *L. pontium*), using a combined morphometric and molecular approach, which is missing in this group.

## 2. Materials and Methods

### 2.1. Plant Material

Field surveys were carried out during the period 2011–2018, along the coast of the Lazio region and in all the islands of the Ponziane Archipelago (central Italy) (Table 1 and Figure 1). All the specimens collected are deposited in the Herbarium RO [32]. In addition, type specimens of the four species were also analyzed: *L. amynclaeum* [Torre Capo Vento, Sperlonga-Gaeta; 8 June 1960, Agostini (RO, isotype)], *L. circaei* [Torre di Paola, Circeo, 7 July 1958, Lusina (RO, holotype)], *L. pontium* [Cala d’Inferno, Ponza Island, 24 August 1959, Muneghina (RO, holotype)], *L. pandatariae* [Montagnozzo, Ventotene Island, 22 September 1901, Béguinot (RO holotype)]. The species nomenclature follows Brullo and Guarino [27].

### 2.2. Molecular Analysis

The molecular regions chosen for this study are located both in the biparental nuclear DNA (nrDNA) and in the uniparental plastid DNA (cpDNA). The plastid genome of most plants is maternally inherited, reflecting gene flow by seeds (e.g., *L. carolinianum* (Walter) Britton analyzed in Corriveau and Coleman [33]); the chosen molecular regions include two introns (*pet*D and *trn*L^(UAA)^) and two intergenic spacers (IGSs) (*pet*B-*pet*D and *trn*L^(UAA)^-*trn*F^(GAA)^), which have already been employed in *Limonium* phylogeography and phylogenies [4,10,23,34,35]. In the nuclear genome, the chosen sequences were the variable fragments of the internal transcribed spacers of ribosomal genes (ITS1 and ITS2, also including the 5.8S gene), which have already been used to study the evolution history of *Limonium* [4,10,35,36,37]. The same populations were also analyzed using a morphometric approach.

### 2.3. Genomic DNA Isolation

In total, 80 individuals were analyzed for a total of ten populations (Table 1). Each population was represented by eight individuals and total genomic DNA was extracted from dried leaves with a GeneAll Exgene Plant SV kit (GeneAll Biotechnology, Seoul, Korea) according to the manufacturer’s instructions. Recalcitrant samples were extracted using a modified CTAB 2X procedure [38]. DNA quality was checked via 1% agarose electrophoresis with SafeView Nucleic Acid Stain (Applied Biological Materials Inc., Richmond, VA, USA) and visualized using the UVIdoc HD5 gel documentation system (UVITEC, Cambridge). DNA concentration was estimated using a Qubit 3 Fluorometer (Invitrogen, Thermo Fisher Scientific, Waltham, MA, USA).

### 2.4. PCR Amplification and Sequence Analyses

ITS marker of the nuclear ribosomal DNA (ITS = ITS1, 5.8S and ITS2) and four plastid markers (*trn*L^(UAA)^ and *pet*D introns, *trn*L^(UAA)^-*trn*F^(GAA)^ and *pet*B-*pet*D IGS) were amplified by using primers reported in the literature. ITS was amplified with JK14 (forward): 5′-GGA GAA GTC GTA ACA AGG TTT CCG-3′ [39] and SN3 (reverse): 5′ -TTC GCT CGC CGT TAC TAA GGG-3′ [40]; *trn*L^(UAA)^ intron plus *trn*L^(UAA)^-*trn*F^(GAA)^ IGS with c (forward) 5′-CGA AAT CGG TAG ACG CTA CG-3′ and f (reverse) 5′-ATT TGA ACT GGT GAC ACG AG-3′ [41]; and *pet*B-*pet*D IGS with *pet*BE2-IGSF (forward) 5′-ATG CAC TTT CCA ATG ATA CG-3′ and *pet*D-E2R (reverse) 5′-CCC GAG GGA ACC GGA CAT-3′ [42]. An internal reverse primer for the *pet*B-*pet*D IGS marker was also designed in this study (Lim_petB-D_373r, 5′-GAA TTC TAT TCA AGC GAA CC-3′).

The volume of each amplification reaction was 20 μL, using ca. 2–4 ng of template DNA, 0.25 μM of each primer, and Phire Plant Direct PCR Master Mix (Thermo Fisher Scientific, Waltham, MA, USA) according to the manufacturer’s instructions. For the recalcitrant or ambiguous samples, high-fidelity Kodaq 2X MasterMix (Applied Biological Materials, Richmond, VA, USA) was used. All amplicons (>500 bp) were purified by PEG 8000 precipitation (PEG 15%, 2.5mM NaCl), with two washes with 80% ethanol, and resuspended in 10 μL of nuclease-free molecular biology grade water (Ambion, Thermo Fisher Scientific, Waltham, MA, USA). Approximately, 7 ng of purified amplicons were sequenced in a volume of 5 μL using 0.5 μL of primer 6.4 μM and fluorescent dyes (Bright Dye Terminator Cycle Sequencing Kit, ICloning). The reactions were purified using a BigDye XTerminator Purification Kit (Applied Biosystems, Thermo Fisher Scientific, Waltham, MA, USA) and electrophoresed on a 3130 Genetic Analyzer (Life Technologies, Thermo Fisher Scientific, Waltham, MA, USA). The sequences’ raw data were analyzed using the AB DNA Sequencing Analysis ver. 5.2 software (Applied Biosystems, Thermo Fisher Scientific, Waltham, MA, USA), then edited and assembled in ChromasPro ver. 2.1.8 software. To verify their identity and check for any contamination, the sequences were compared with the public database nucleotide collection using the BLASTn (https://blast.ncbi.nlm.nih.gov (accessed on 15 January 2022). Sequences obtained in this study have been deposited in GenBank under accession numbers OP452889–OP452889–OP452890–OP452891–OP452892 (ITS), OP485326–OP485327–OP485328–OP485329 (*pe*tB-*pet*D IGS + *pet*D intron), OP485330–OP485331–OP485332–OP485333 (*trn*L^(UAA)^*-trn*F^(GAA)^ IGS) (Table 2 and Table 3).

ITS amplicons with multiple peaks within the sequence were cloned using the CloneJET PCR Cloning Kit (Thermo Fisher Scientific, Waltham, MA, USA) according to the manufacturer’s instructions. Transformation was performed using StrataClone SoloPack Competent Cells (Agilent Technologies, Santa Clara, CA, USA). The bacteria were cultured in LB medium at 37 °C for 30 min and subsequently transferred to LB agar plates containing 100 ug/mL ampicillin ON. Thirty-two randomly selected clones from each transformation were amplified using the corresponding PCR primers and ten amplicons were sequenced.

### 2.5. Data Analysis

Sequences datasets from nrDNA (ITS) and cpDNA (*pet*B-*pet*D IGS+*pet*D intron and *trn*L^(UAA)^ intron + *trn*L^(UAA)^-*trn*F^(GAA)^ IGS) were analyzed separately. The sequences were aligned with Clustal W [43] as implemented in the BioEdit ver. 7.2.5 software package [44] and checked manually. A geographic map was generated to assess the distribution and frequency both genotypes (nrDNA) and haplotypes (cpDNA) present in the ten *Limonium* populations in this study.

*NrDNA sequences*. To check the phylogenetic collocation of the *Limonium* accessions in this study, a Bayesian inference (BI) framework was performed with MrBayes ver. 3.2.6 software [45] and using the dataset of Malekmohammadi et al. [4] which was kindly provided by Myriam Malekmohammadi. Some ITS accessions from Koutroumpa et al. and Thornhill et al. [10,46] were used because both belong to the *L. multiforme* complex (i.e., *L. remostispiculum* (Lacaita) Pignatti) and have a high sequence identity from BLASTn analyses (cut-off >99%; see results) (Appendix A). From the Malekmohammadi et al. dataset, the ITS sequences that had more than 50 contiguous basepairs missing were discarded from the analysis (i.e., *L. australe* Kuntze, *L. aureum* (L.) Chaz., *L. haitiense* S.F.Blake, *L. mucronulatum* (H. Lindb.) Greuter & Burdet, *L. pectinatum* (Aiton) Kuntze, *L. pyramidatum* Brullo & Erben, and *L. sundingii* Leyens, Lobin, N.Kilian & Erben). *Plumbago auriculata* Lam. was used as an outgroup as also reported in Malekmohammadi et al. [4]. The most likely substitution model for the ITS marker was computed by using the jModeltest ver. 2.1.10 software [47] and the GTR + G + I model was the better model according to the Akaike information criterion (AIC). Two runs of four Markov chains (three hot, one cold) were performed for 10,000,000 generations, sampling every 1000 generations, and discarding the first 20% as burn-in. Convergence diagnostics were also checked with Tracer ver. 1.7.1 software [48]. A maximum likelihood (ML) tree search was performed using RaxML-NG via its web server portal (https://raxml-ng.vital-it.ch/#/; (accessed on 18 April 2022) [49]). Bootstrap analyses were carried out with an automatic number of replicates with a bootstopping cutoff of 0.03.

*CpDNA sequences*. A preliminary phylogenetic analysis (BI and ML inference) was performed on combined plastid matrix (i.e., *pet*B-D+ *trn*L^(UAA)^-F^(GAA)^ regions) with literature data to evaluate the haplotypes’ relationship. We used the same alignment data file employed in Malekmohammadi et al. [4], which was kindly provided by M. Malekmohammadi (Appendix A). From this reference dataset, the plastid sequences that had more than 100 contiguous basepairs missing were discarded from the analysis (i.e., *L. anatolicum* Hedge, *L. brasiliense* (Boiss.) Kuntze, *L. carolinianum* Britton, *L. michelsonii* Lincz., *L. sarcophyllum* Ghaz. & J.R.Edm., *L. vigaroense* Marrero Rodr. & R.S.Almeida, and *Myriolimon diffusum* (Pourr.) Lledó, Erben & M.B.Crespo). Mutational hotspots (including poly-A/T stretches) with ambiguous homology assessment were excluded from analysis as performed in Malekmohammadi et al. [4]. *Plumbago auriculata* was used as an outgroup as also reported in Malekmohammadi et al. [4]. The most likely substitution model for both markers was GTR + I and the same BI and ML settings used in nuclear data were used for these analyses.

If our haplotypes’ relationships were not evident in phylogeny inference due to a low non-discriminating variability among them, the assessment of haplotypes’ relationships was conducted using TCS version 1.21 software [50] with the optimality criterion of maximum parsimony [51] and treating the gaps as a fifth state. Mononucleotide repeats (polyN) from the cpDNA sequences were included in the analysis, if fixed in the populations sampled (i.e., no variation among individuals/population). Indels were considered as a single mutation event and were therefore coded as single positions in the final alignment. TCS was run with a default parsimony connection limit of 95% among *Limonium* sample data and a connection limit to 45 steps using the outgroup. *Limonium pruinosum* (L.) Chaz. was used as an outgroup according to both Malekmohammadi et al. [4] and our preliminary phylogeny (GenBank accession numbers *pet*B-*pet*D IGS+ *pet*D intron, MF083795; *trn*L^(UAA)^ intron+*trn*L^(UAA)^-*trn*F^(GAA)^, MF083894).

### 2.6. Morphometric Analysis

Ten populations (Figure 1), 25 individuals for each one, plus the holotypes of the four species considered in the research were investigated (Table 1). Twenty-two quantitative characters (3 discrete, 19 continuous (Table 4); see Figure 2 for the details of the inflorescence) were measured in 254 individuals (a total of 5742 measurements) using a Zeiss GXS stereomicroscope. The data matrix (individuals × variables) was processed using the software package NCSS 2007. The variability of the characters was examined by principal component analysis (PCA), discriminant analysis (DA), and box plots. DA was performed using the first six components derived from PCA, which explain about the 70% of the total variability. The use of component scores (each linearly independent by construction) allowed an unbiased discriminant model both solving the indeterminacy due to multicollinearity of the independent variables and providing a more reliable prediction for the smaller number of involved variables [52,53,54]. As a supervised technique, we performed the DA on groups classified both as species and as localities. The matrix of actual/predicted groups was analyzed by comparing the values among these groups, especially regarding the diagonal, whose values reveal the matching of actual and predicted observations for each group. The value of correct classification reported in the results is the classification accuracy achieved by the actual discriminant functions over what is expected. A multivariate analysis of variance (MANOVA) was also performed to test the significance of differences between response (dependent) variables (morphological characters) and factor variables (=groups, i.e., taxa). Relevant literature (including protologues; [28,29]) was also analyzed.

## 3. Results

### 3.1. Molecular Analyses

*Nuclear data.* The ITS of our dataset is from 620 to 622 characters. Several specimens belonging to SS (4), AR (7), ST (4), TE (6) populations, and two type specimens (*L. circaei* and *L. pandatariae*) presented ITSs with sequences with double peaks caused by indels and two single different nucleotides (Table 2 and Figure 3). After cloning and sequencing, in total four single ITS genotypes were identified (blue—N1 (620 bp), green—N2 (621 bp), pink—N3 (620 bp), and brown—N4 (622 bp); Table 2) which determined three “mixed patterns” (orange—H1, white—H2, and yellow—H3) as shown in Figure 3. In the rest of the dataset, two single ITS genotypes were observed (N1 and N2), of which N1 is present with a greater frequency in all populations (98.8 vs. 19%, respectively) as a single genotype or as a mixed pattern; in fact, only one specimen from the population AR does not present the N1 genotype (Figure 3). The H3 mixed pattern was exclusive to the TE population as was the presence of the single genotype N2 in the AR population. In this population, the H1 mixed pattern was also present with higher frequency. The H3 mixed pattern was exclusive to the TE population as well as the H2 mixed pattern which was observed only in the type specimens of *L. circaei* (Figure 3).

After the BLASTn analyses, the identity of the single ITS genotypes was N1 = 100% (query cover 95%) with *L. circaei* (GenBank: MH582583), *L. carthaginense* (Rouy) C.E.Hubb. & Sandwith (MH582582), *L. corsicum* Erben (MH582581), and *L. bonifaciense* Arrigoni & Diana (MH582580); N2 = 99.19% (q.c. 93%) with *L. cumanum* (Ten.) Kuntze (JX983717) and 99.05% (q.c. 95%) with *L. saracinatum* R.Artelari (MH582600); N3 = 100% (q.c. 95%) with *L. multiforme* Pignatti (MH582584); and N4 = 100% (q.c. 100%) with *L. duriusculum* (Girard) Fourr. p.p. (MF963815), (q.c. 96%) *L. densissimum* (Pignatti) Pignatti (MH582610), (q.c. 95%) *L. gueneri* Dogan, Duman & Akaydı (MF041873), and (q.c. 95%) off-spring of *L. ovalifolium* (Poir.) Kuntze x *L. nydeggeri* Erben (MK005224).

The length of the ITS sequence alignment with the 100 accessions from the literature was 740 characters, 450 of which were variable and 354 of which were parsimony informative (Appendix A). Considering only our single genotypes, their alignment had 13 variable sites and one parsimony informative site. According to the ML and BI phylogenetic analyses (Figure 3), our four genotypes fell into the “*L. graecum* clade” which corresponds to node F of Figure 3 of Malekmohammadi et al. [4]. The N1 genotype was sister to the N3 genotype, and these genotypes are related to the N2 genotype. All genotypes fell into the group of accessions of the taxa afferent to the *L. multiforme* group (i.e., N1 with *L. circaei* with a sequence identity of 100%, N2 was sister to *L. cumanum* and *L. remotispiculum*, and N3 with two accessions of *L. multiforme*, of which one shares a 100% sequence identity). The N4 genotype was present in a different unresolved group consisting of collapsed taxa not belonging to the *L. multiforme* group. All our genotypes’ accessions are well supported according to BI while, on the contrary, in ML analyses, only the N4 genotype had maximum support (Figure 3).

*Plastid data*. Excluding the primer and about the first 25 bases due to reading by the automated sequencer, the length of the *trn*L^(UAA)^ intron together with *trn*L^(UAA)^-*trn*F^(GAA)^ IGS was between 897 and 898 bp because of a variable polyA in the *trn*L^(UAA)^ intron (Table 3). No SNPs were detected except for polyA which was A5 in western island populations (codes IF, SS, ZA; Figure 4) and A6 in the rest of the dataset. By BLASTn analyses, this marker (excluding polyA) was not able to discriminate among the *Limonium* taxa in the study, and more than 20 species from the GenBank database share 100% identity with our sequences (e.g., *L. circaei*, *L. bonifaciense*, and *L. saracinatum* R.Artelari).

The *pet*B-*pet*D IGS plus *pet*D intron was 868 bp in length and three variable sites that were not parsimony informative (one in the IGS and two in the intron) were observed determining four haplotypes (Table 3). By BLASTn analyses, these haplotypes present a sequence identity of 99.77–99.88% (q.c. 100%) from two to three *Limonium* taxa present in the GenBank database (R = 99.77% *L. connivens* Erben, *L. cumanum*, and *L. multiforme*; G = 99.77%, *L. connivens* and *L. multiforme*; B = 99.77%, *L. connivens* and *L. multiforme*; A = 99.88%, *L. connivens* and *L. multiforme*). By combining the sequences of all plastid markers, four haplotypes were always observed with a geographical distribution as previously observed (orange—A, blue—B, green—G, and red—R; Table 3 and Figure 4). Two haplotypes were exclusive to island populations (red—R and blue—B for western and eastern island populations, respectively). The green (G) haplotype was present in all continental populations except for *L. circaei* type and was very rare in island populations where it was observed in the two type specimens (*L. pandatariae* and *L. pontium*) and in one sample of the AR population; a unique haplotype (orange—A) is present for the type of *L. circaei* as also observed for ITS data (Figure 3 and Figure 4).

The length of plastid sequence alignment matrix with 94 accessions from Malekmohammadi et al. [4] was 1898 characters (474 of which were variable and 275 of which were parsimony informative; Appendix A). According to the ML and BI phylogeny inference, our plastid accessions are always placed in the “*L. graecum* clade” of Figure 2 (node F) in Malekmohammadi et al. [4]. In our analyses, they formed a supported single group (99/1; bootstrap/posterior probability, respectively) sister to *L. virgatum* and successively to several collapsed accessions as *L. cumanum*/*L. multiforme*/*L. connivens*/*L. gueneri*/*L. cumanum*/*L. gibertii* (Appendix A). Considering the TCS analyses, the ancestral haplotype was A (orange in Figure 4) which corresponds to the type specimen of *L. circaei* from which the other three haplotypes derive independently and presents a geographical correlation as previously reported (Figure 4).

### 3.2. Morphometric Analysis

The PCA of the 22 morphological characters analyzed (Table 4) shows that the cumulative percentage of eigenvalues for the first seven axes is 10.17%, with a higher contribution (more than 10%) given by the first three components (23.24%, 13.83%, and 9.82%, respectively). The examination of the combined graphs among pairs of these seven components shows three separated groups along the first and second components (Figure 5). These groups correspond to the populations from (1) TE (Terracina), (2) AR (Ventotene), and (3) a large group including the remaining localities. The highest contributions to axes were given by the following characters: length of the leaves, number of fertile and sterile branches, average angle between branches, length of the ludicule, number of flowers, length of the outer, median, and inner bracts, width of the median and outer bract, length of limb, and width of the corolla lobes.

The DA shows different results depending on the use of the species names or the localities to classify the groups:(1)When we classified the populations using the localities’ names (= ten groups; see Table 1), DA predicted four groups (Figure 6) based on the first two discriminant functions which explain 72.2% of the total variation (eigenvalues: 40.5% (1st function) and 31.7% (2nd function)). These four groups, partially overlapped, correspond to the following localities: A) AR (Ventotene), B) CV (Sperlonga), C) TE (Terracina), and D) a large group including the remaining localities. The matrix of actual/predicted groups displays high percentages along the diagonal (whose values reveal the matching of actual and predicted observations for each group) for CV (100%), TE (96%), and AR (82%), whereas low or very low percentages characterized the diagonal values of the other groups. The value of correct classification is low (61.8%).(2)When we classified the populations using the names of the species (= four groups, *L. amynclaeum*, *L. circaei*, *L. pandatariae*, and *L. pontium*), DA predicted two partially overlapped groups (Figure 7) based on the first two discriminant functions which explain 84.5% of the total variation (eigenvalues: 64.9% (1st function) and 19.6% (2nd function)). These two groups correspond to (A) *L. pantadariae* (one locality, AR (Ventotene); see Table 1) and (B) a group comprising all the individuals identified as *L. amynclaeum*, *L. circaei*, and *L. pontium* and collected in the remaining nine localities (coast of Lazio and the islands of the Ponziane archipelago except Ventotene; see Table 1). In fact, the matrix of actual/predicted groups displays a percentage of the diagonal value of *L. pandatariae* of 56%. The value of correct classification is low (57.5%).

**Figure 6 plants-11-03163-f006:**
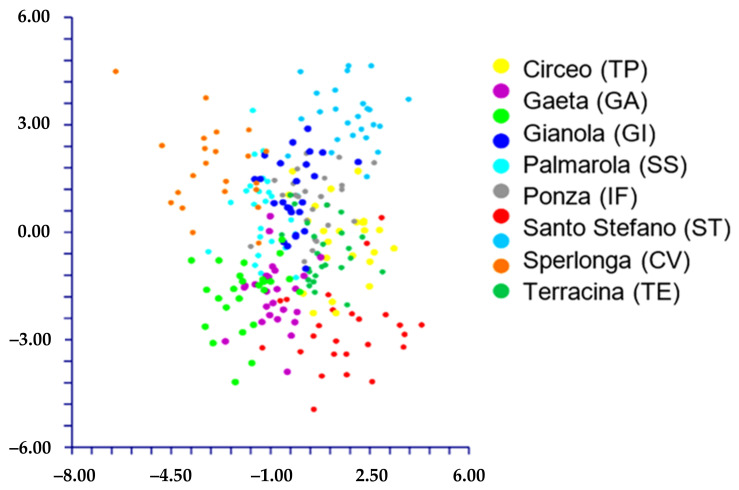
DA (first (*x* axis) vs. second (*y* axis) components) performed on groups classified using the localities’ names.

**Figure 7 plants-11-03163-f007:**
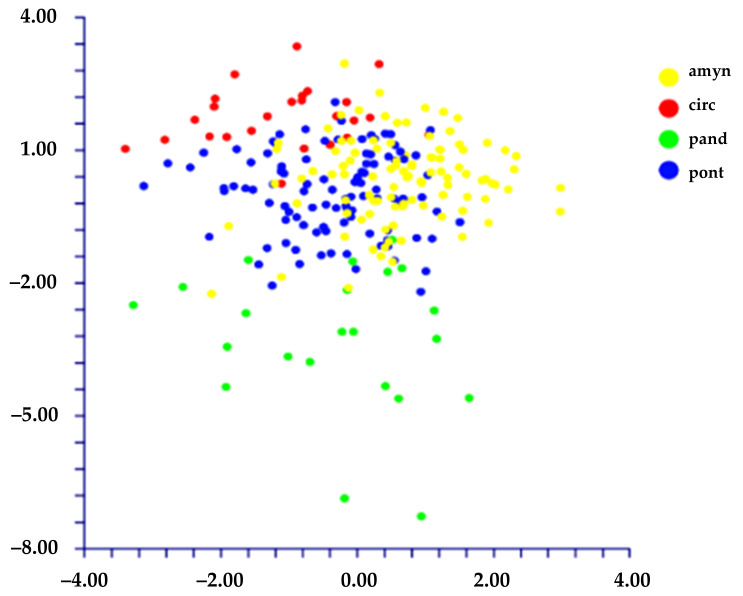
DA (first (*x* axis) vs. second (*y* axis) components) performed on groups classified using the species names (amyn = *L. amynclaeum*; circ = *L. circaei*; pand = *L. pandatariae*; pont = *L. pontium*).

Furthermore, we performed the DA using the three groups generated from the PCA (*L. pandatariae* from AR (Ventotene), *Limonium* sp. from TE (Terracina), and *L. pontium* (including *L. amynclaeum* and *L. circaei*) from all other localities). The result is that these three groups are statistically well supported, based on the two discriminant functions which explain 100% of the total variation (eigenvalues: 52.1% (1st function), and 47.9% (2nd function)) (Figure 8). The value of correct classification is high (86.4%).

Finally, the box plots, made using the characters derived from PCA that are able to discriminate the various groups (length of the leaves, number of fertile and sterile branches, average angle between branches, length of the ludicule, number of flowers per spikelet, length of the outer, middle, and inner bracts, width of the middle and inner bract, length of limb, and width of the calyx lobes), confirm the separation of the following three groups: AR (Ventotene), TE (Terracina), and all other localities together (Figure 9).

The results of the MANOVA show significant differences at both species and population levels. Probability level is less than 0.000001 for all the statistical tests considered (Wilks’ lambda, Hotelling–Lawley trace, Pillai’s trace, and Roy’s largest root). F-ratios are high, ranging from F = 16.35 to 17.97 (Table 5).

## 4. Discussion

According to the molecular data, the information obtained has been useful to give another additional reading key to the morphometry approach on these taxa belonging to the *L. multiforme* group. Analyzing the ITS (nrDNA), which is biparental inheritance, the presence of mixed genotypes suggests that sexuality events have probably been or still are recurrent in these taxa as supposed in Brullo and Guarino [27]. The presence of the N1 genotype (Figure 3) was always observed in all the individuals examined (except one individual of the AR population) which had either single or mixed genotypes. The other single genotypes observed (Figure 3) are close to taxa belonging to the *L. multiforme* group (i.e., see phylogenetic position of N1, N2, and N3 genotypes, Figure 3). This confirms how this group is a set of related taxa. For example, the population from the island of Ventotene (code AR and locus classicus of *L. pandatariae*, Table 1) presents ITS genotypes related to *L. circaei*, *L. cumanum*, and *L. remotispiculum* (Figure 3), whereas the population from the Torre Paola locality (code TP and locus classicus of *L. circaei*, Table 1) is related to *L. circaei* and *L. multiforme* (Figure 3). On the other hand, the population from Terracina (code TE, Table 1), in addition to having the ubiquitous N1 genotype, presents a very different genotype that has a phylogenetic affinity to species that do not belong to the *L. multiforme* group (Figure 3).

Concerning the analyses of uniparental plastid markers (cpDNA) (Figure 4), it has been possible to deduce both (1) a geographical correlation of the distribution of the haplotypes which all derive from an ancestral form like the type of *L. circaei*; and (2) that several colonization events from the mainland (Tyrrhenian coast) and neighboring islands (Ponziane Archipelago) have occurred over time among *Limonium* populations (e.g., see green haplotype (code G) in Figure 4). In fact, the long seed dispersion by wind, birds, and water (along the coast) of *Limonium* is present [34,55] and the fruits can float for a long time in seawater without destruction of their germination power as observed in *L. vulgare* Mill. by Koutstaal et al. [55] and *L. ramosissimum* (Poir.) Maire subsp. *provinciale* (Pignatti) Pignatti [56]. So, if with a molecular approach we have given a broader interpretation key by also integrating the data with the published phylogenies [4,10,35,36,37], we are reasonably in agreement with what is reported in Brullo and Guarino [27] for the taxa belonging to the *L. multiforme* group. In fact, the taxa analyzed would not show apomixis (e.g., for shared molecular markers, Figure 3 and Figure 4), they might have been (or are) interfertile (e.g., due to the presence of different shared ITS genotypes, Figure 3), and the geographical isolation (e.g., see haplotypes distribution, Figure 4) has been and still is of fundamental importance to determine the variability that has been detected from the morphological point of view.

Morphometric data obtained highlight that three groups are statistically well supported, and they are correlated with geographical distribution areas, i.e., populations from (1) Ventotene (the southmost island of the Ponziane Archipelago), (2) Terracina, and (3) the all other localities together. Five to seven characters allow us to distinguish these groups. According to the literature, *Limonium* species were often distinguished from each other based on two to five morphological characters [see, e.g., ([3,57] (35 new described species), 2 (35 new described species), [12,58])].

All things considered, we here propose to consider all populations studied as part of a single species whose name is to be *Limonium pontium* due to the nomenclatural priority over the other available names (Art. 11.4 of ICN; [59]). Three subspecies are here recognized (see the taxonomic treatment below): subsp. *pontium* (populations occurring along the coast from Circeo to Gianola localities (excluding Terracina) and the other ones from the islands of Ponza, Palmarola, Zannone, and Santo Stefano), subsp. *pandatariae* (Pignatti) Iamonico, Iberite, De Castro & Nicolella, *comb. et stat*. *nov.* (populations from the island of Ventotene), and subsp. *terracinense* Iberite, Iamonico, De Castro & Nicolella, *subsp. nov.* (no name is available for Terracina’s population). They are supported by both molecular and morphometric data. Concerning *L. circaei*, it is also synonymized here with *L. pontium* s.s. since neither molecular nor morphological differences exist for the studied populations. Finally, the Sperlonga population (previously named as *L. amynclaeum*) can be distinguished from the morphological point of view but there is no molecular support. Consequently, we here synonymize *L. amynclaeum* with *L. pontium* subsp. *pontium*.

## 5. Taxonomic Treatment

***Limonium pontium*** Pignatti, Bot. J. Linn. Soc. 64: 264. 1971 subsp. ***pontium***. Holotype: Italy, Lazio region, Ponza Island, Cala d’Inferno, 24.08.1959, A. Muneghina s.n. (RO!).

= *Limonium circaei* Pignatti, Webbia 36: 50. 1982, *syn. nov.* Holotype: Italy, Lazio region, Torre di Paola, Circeo, 07.07.1958, G. Lusina s.n. (RO!).

= *Limonium amyclaeum* Pignatti, Webbia 36: 49. 1982, *syn. nov.* Holotype: Italy, Lazio region, Torre Capo Vento, Sperlonga-Gaeta, 08.06.1960, Agostini s.n. (TSB!, isotype RO!).

**Description:** plant perennial, glabrous, forming a subshrub (50–)115–190(–427) cm tall, branched, average angle between branches (35–)54–70(–106)°. Leaves only in basal rosettes, green-greyish, verrucose, with revolute border, (7–)14.2–29.3(–50) mm long and (1–)2.8–5.0(–10) cm broad, spatulate, apex rounded or slightly retuse, base attenuate, not mucronate, with 1 conspicuous central nerve (later nerves not visible). Inflorescence in panicles of spikes; each spike is composed of spikelets. Sterile branches (0–)6–19(–76), branched or not. Fertile branches (4–)8–21(–163). Spikes (2–)8.7–17.8(–40) mm long, usually curved. Ludicule (0.5–)1.3–2.1(–4.3) mm. Spikelets (3.2–)5.0–5.9(–9.7) mm long, each one with 1–2(–3) flowers. Outer bract (0.6–)1.0–1.3(–2.5) mm long and 1.5-2.0 mm broad, triangular-ovate, acute, with margin slightly membranous. Middle bract (0.8–)1.4–1.9(–3.3) mm long and (0.3–)0.7–1.1(–1.7) mm broad, rhombic, acute with excurring median nerve and margin broadly membranous. Inner bract (2.8–)3.5–4.2(–4.9) mm long and (0.8–)1.5–1(–2.9) mm broad, elliptic, rounded, with margin broadly membranous. Calyx with 5 ribs, each one with sparse hairs at the proximal part; tube (0.7–)1.5–2.1(–2.6) mm long and campanulate limb (1.4–)2.3–3.0(–4.1) mm long; calyx lobes (0.3–)0.6–1.0(–1.3) mm long, (0.2–)0.50–0.75(–1.0) mm broad. Corolla 4.3–6.4 mm long, lilac.

**Etymology:** The specific epithet derives from the ancient name of the island of Ponza, i.e., “Pontia”.

**Distribution and habitat:** An endemic subspecies growing on calcareous cliffs along the coast from Circeo to Gianola localities (excluding Terracina) and on volcanic cliffs on the islands of Ponza, Palmarola, Zannone, and Santo Stefano.

***Limonium pontium*** Pignatti, Bot. J. Linn. Soc. 64: 264. 1971 subsp. ***terracinense*** Iberite, Iamonico, De Castro & Nicolella, *subsp. nov.* Holotype: Italy, Lazio region, Terracina, Torre Gregoriana, 13.07.2015, M. Iberite & G. Nicolella s.n. (RO!, isotype FI!).

**Diagnosis:***Limonium pontium* subsp. *terracinense* differs from *L. pontium* subsp. *pontium* and *L. pontium* subsp. *pandatariae* in having (1) smaller average angle between branches ((28–)47–54(–64) vs. (35–)54–70(106)° (subsp. *pontium*)); (2) longer ludicule ((1.7–)2.6–3.4(–5.6) vs. (0.5–)1.3–2.1(–4.3) (subsp. *pontium*)); (3) longer bracts (outers: (1.5–)1.9–2.2(–2.5) mm vs. (0.6–)1.0–1.3(–2.5) (subsp. *pontium*) and (0.5–)1.1–1.7(–1.9) (subsp. *pandatariae*); inners: (3.4–)4.6–5.0(–6.1) mm vs. (2.8–)3.5–4.2(–4.9) (subsp. *pontium*) and (3.2–)3.7–4.1(–5.0) (subsp. *pandatariae*)); (4) wider middle and inner bracts (middle: (0.8–)1.2–1.6(–1.8) vs. (0.3–)0.7–1.0(–1.7) (subsp. *pontium*) and (0.4–)0.5–0.8(–1.1) (subsp. *pandatariae*); inner: (1.6–)1.9–2.8(–2.9) vs. (0.8–)1.5–1.0(–2.9) (subsp. *pontium*) and (1.4–)1.7–2.0(–4.0) (subsp. *pandatariae*)); (5) higher number of flowers per spikelet ((1–)2–3(–4) vs. 1–2(–3) (subsp. *pontium*) and 1–2 (subsp. *pandatariae*)); (6) longer limb ((2.1–)3.0–3.5(–3.9) vs. (1.4–)2.3–3.0(–4.1) (subsp. *pontium*) and (0.5–)2.2–3.0(–3.3) (subsp. *pandatariae*)); (7) wider lobes ((0.5–)0.85–1.15(–1.1) vs. (0.2–)0.50–0.75(–1.0) (subsp. *pontium*) and (0.4–)0.55–0.65(–1.0) (subsp. *pandatariae*)).

**Etymology:** The subspecific epithet derives from one of the ancient Roman names of the locus classicus of the taxon, i.e., “Terracina”.

**Proposed vernacular names:** Sea lavender of Terracina (English), Limonio di Terracina (Italian).

**Distribution and habitat:** An endemic subspecies occurring only in the locus classicus (Terracina) on calcareous cliffs.

***Limonium pontium*** Pignatti, Bot. J. Linn. Soc. 64: 264. 1971 subsp. ***pandatariae*** (Pignatti) Iamonico, Iberite, De Castro & Nicolella, *comb. et stat. nov.* Bas.: *Limonium pandatariae* Pignatti, Webbia 36(1): 54 1982—holotype: Italy, Lazio region, island of Ventotene, Montagnozzo, 22.09.1901, A. Béguinot s.n. (RO!).

**Diagnosis:***Limonium pontium* subsp. *pandatariae* differs from *L. pontium* subsp. *pontium* and *L. pontium* subsp. *terracinense* in having (1) smaller average angle between branches ((41–)49–54(–62) vs. (35–)54–70(106)° (subsp. *pontium*)); (2) higher number of sterile branches ((9–)22–44(–92) vs. (0–)6–19(–76) (subsp. *pontium*) and (3–)6–9(–44) (subsp. *terracinense*)); (3) higher number of fertile branches ((8–)19–58(–94) vs. (4–)8–21(–163) (subsp. *pontium*) and (5–)6–22(–59) (subsp. *terracinense*)); and (just from subsp. *pontium*) by (4) longer leaves ((22–)30.5–48.0(–75) vs. (7–)14.2–29.3(–50) (subsp. *pontium*)); (5) longer ludicule ((2.1–)2.6–4.2(–9.2) vs. (0.5–)1.3–2.1(–4.3)); and (6) smaller middle bracts (length: (0.7–)1.2–1.4(–1.8) vs. (0.8–)1.4–1.9(–3.3) (subsp. *pontium*); width: (0.4–)0.55–0.79(–1.1) vs. (0.3–)0.70–1.10(–1.7) (subsp. *pontium*)).

**Etymology:** The subspecific epithet derives from the ancient name of the island of Ventotene, i.e., “Pandataria”.

**Distribution and habitat:** An endemic subspecies occurring on the island of Ventotene, on volcanic cliffs.

### Diagnostic Key

On the basis of the results obtained, we propose a corrected diagnostic key (from step no. 9 onwards) given by Ref. [26] for the complex of *Limonium multiforme*.

9. Leaves 3.6–5.3 mm wide……………………………………………………………………………………………… 1010. Spikelets 1–2 per cm; inner bract 3.0–3.5 mm long…………… *Limonium**remotispiculum*10. Spikelets 2–6 per cm; inner bract 3.7–4.3 mm long……………………………………… 1111. Spikes 25–80 mm long; outer bract 1.5–2.0 mm long………… *Limonium brutium* Brullo11. Spikes 8–18 mm long; outer bract 1.1–1.6 mm long ……………
*Limonium pontium* s.lat.11a. Flowers per spikelet (1–)2–3(–4); outer bract 1.8–2.1 mm long; inner bract 4.5–5.1 mm long; limb 1.1–1.6 mm long ………………………………………………… subsp. *terracinense*11b. Flowers per spikelet 1–2(–3); outer bract 1.0–1.6 mm
long; inner bract 3.5–4.2 mm long; limb 0.7–1.1 mm long………………………………………....................................... 1212a. Leaves 14.2–29.3 mm long; average angle between branches
49–54°; sterile branches 22–44; fertile branches 19–58; middle bracts 1.2–1.4
× 0.5–0.8…………………………………………………………................................ subsp.
*pandatariae*12b. Leaves 30.5–48.0 mm long; average angle between branches
54–70°; sterile branches 6–19; fertile branches 8–21; middle bracts 1.4–1.9 ×
0.7–1.1………………………………………………………………………………… subsp.*pontium*9. Leaves 5.0–11.0 mm wide…………………………………………………………………… 1313. Leaves flat, densely arranged along the caudice
spikelets 2–3 per  cm…………………………………………………………………… *Limonium gorgonae* Pignatti13. Leaves with margin revolute to convolute laxly arranged along the caudice; spikelets 3–4 per cm ………………………………………………………………… *Limonium*
*multiforme*

## Figures and Tables

**Figure 1 plants-11-03163-f001:**
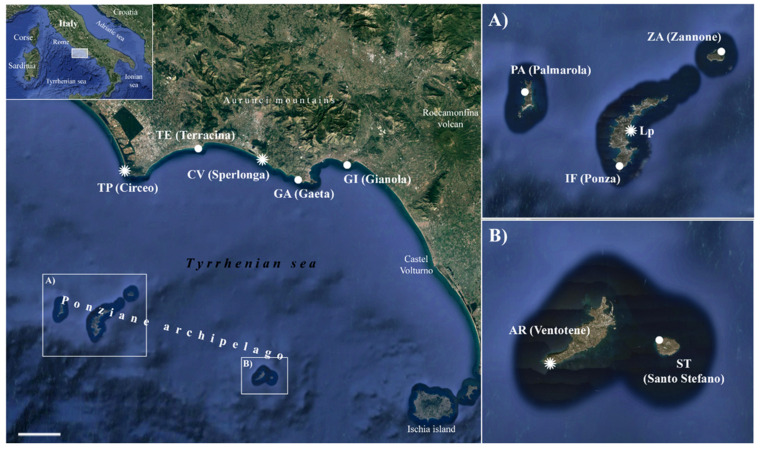
Map of the *Limonium* populations studied; detailed maps of islands (**A**) Palmarola, Ponza, and Zannone, and (**B**) Ventotene and Santo Stefano. Codes follow Table 1.

**Figure 2 plants-11-03163-f002:**
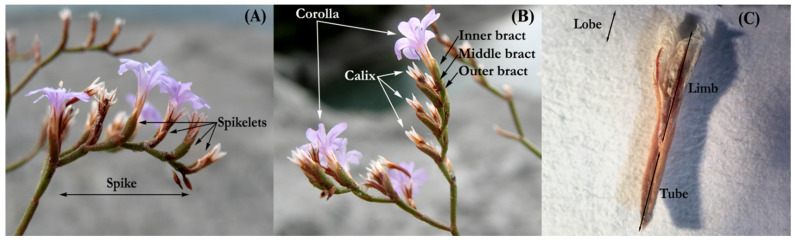
Inflorescence architecture (**A**) and floral parts (**B**,**C**). Picture refers to *Limonium pontium s.str*.

**Figure 3 plants-11-03163-f003:**
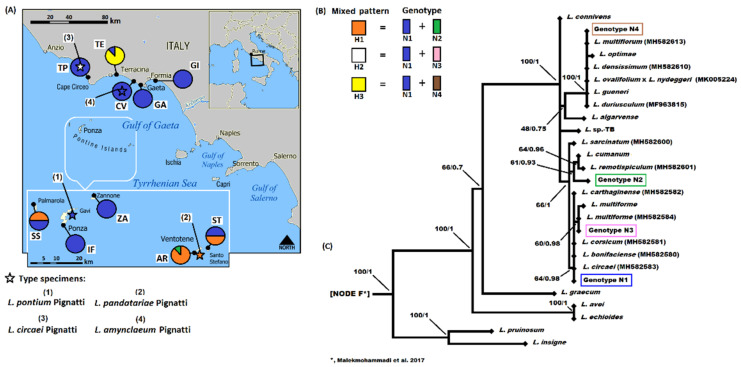
(**A**) The ten localities of the investigated Limonium populations (80 individuals) and the corresponding genotypes (single and mixed) obtained with the ITS markers (nrDNA). Type specimens for four Limonium species analyzed are reported as star symbols with numeric legend. (**B**) Information is also shown for their genotypes and (**C**) corresponding phylogenetic position in the maximum likelihood (ML) tree (a detail of phylogram is shown). Same topology was observed in the Bayesian inference (BI). The dataset of Malekmohammadi et al. [4] has been used in the phylogenetic analysis, the taxa with GenBank numbers in parentheses do not belong to the Malekmohammadi et al. dataset. Node F corresponds to the node of “*L. graecum* clade” present in Figure 3 of Malekmohammadi et al. [4]. ML bootstrap values followed by the Bayesian posterior probabilities are shown below the branches (values >50%). Codes correspond to the population localities shown in Table 1; colored symbols correspond to the genotypes (N1–N3) shown in Table 2.

**Figure 4 plants-11-03163-f004:**
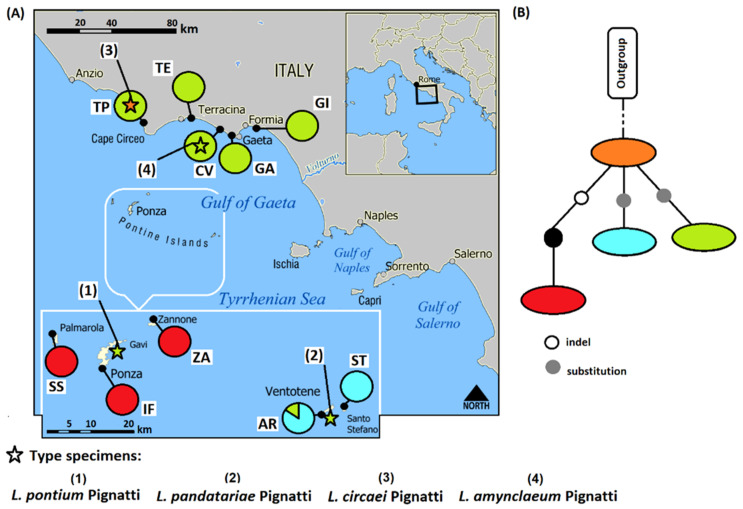
(**A**) The ten localities of the investigated *Limonium* populations (80 individuals) and corresponding haplotypes obtained with *trn*L^(UAA)^ intron + *trn*L^(UAA)^-*trn*F^(GAA)^ IGS and *pet*B-*pet*D IGS + *pet*D intron (cpDNA). Type specimens for the four *Limonium* species analyzed were reported as star symbols with numeric legend. (**B**) Information is also shown for the plastid DNA haplotypes and the corresponding statistical parsimony network using TCS software where the black dots represent undetected haplotypes (outgroup = *L. pruinosum*). Codes correspond to the population localities shown in Table 1; colored symbols correspond to the haplotypes shown in Table 3.

**Figure 5 plants-11-03163-f005:**
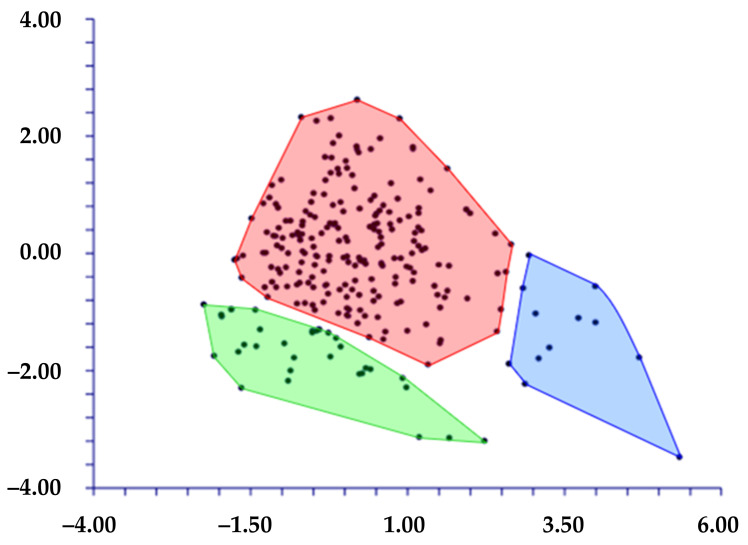
PCA (first (*x* axis) vs. second (*y* axis) components) based on the 22 quantitative morphological characters. Blue polygon: Ventotene population; green polygon: Terracina population; red polygon: other populations.

**Figure 8 plants-11-03163-f008:**
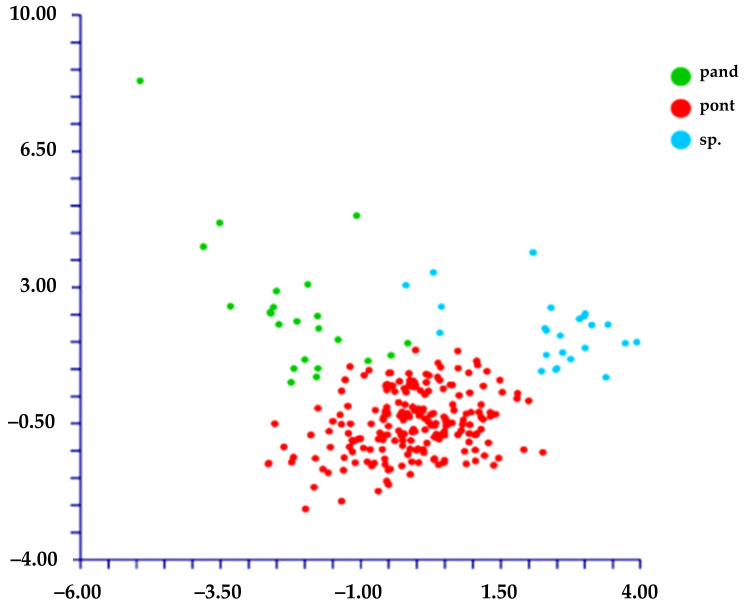
DA [first (*x* axis) vs. second (*y* axis) components) performed on groups derived from PCA (pand = *L. pandatariae*; pont = *L. pontium* (including *L. circaei* and *L. amynclaeum*); sp. = Terracina population).

**Figure 9 plants-11-03163-f009:**
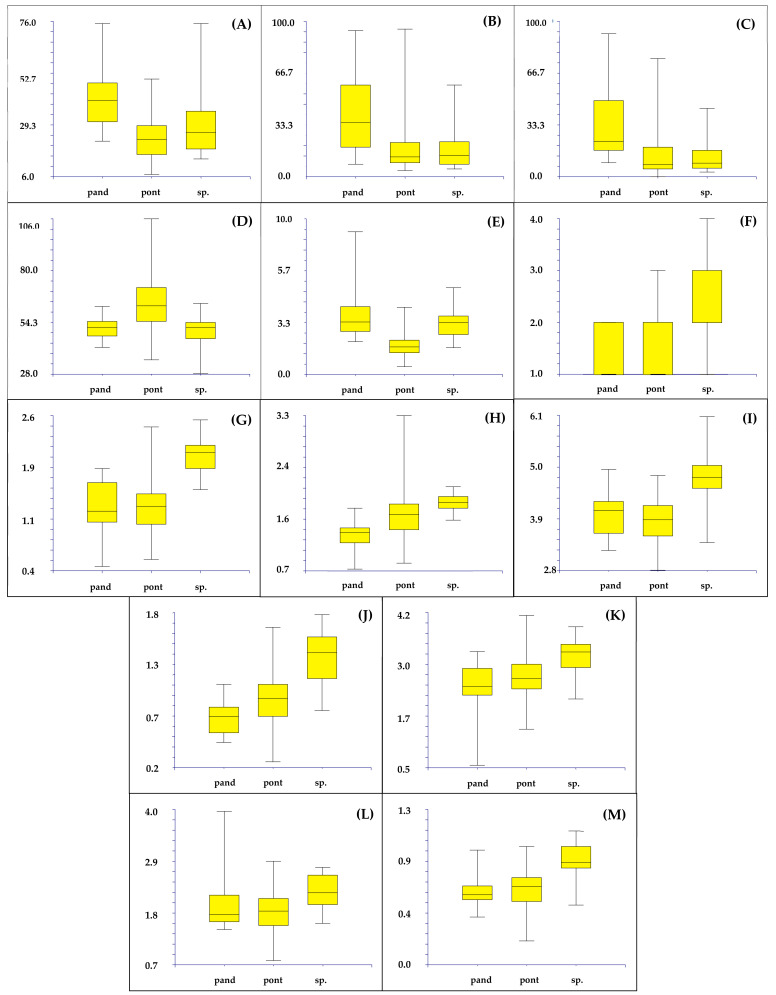
Box plots illustrating the variability of the diagnostic character derived from PCA: length of the leaves (**A**), number of fertile (**B**) and sterile (**C**) branches, average angle between branches (**D**), length of the ludicule (**E**), number of flowers per spikelet (**F**), length of the outer (**G**), middle (**H**), and inner (**I**) bracts, width of the middle (**J**) and inner bracts (**K**), length of limb (**L**), and width of the calyx lobes (**M**) (measurements are in mm). pand = *L. pandatariae*; pont = *L. pontium*; sp. = Terracina population. Yellow boxes illustrate interquartile ranges (=the range between the 25th and 75th percentile) and medians (horizontal line); vertical lines are the whiskers which represent the scores outside the middle 50% (i.e., the lower 25% of scores and the upper 25% of scores).

**Table 1 plants-11-03163-t001:** Populations studied. Asterisk (*) indicates the locus classicus. Species names according to Brullo and Guarino [27].

Species Name	Population Code	Locality
*Limonium amynclaeum* Pignatti	CV	Lazio region, Sperlonga, Torre Capovento *, 13°28′27.54″ E, 41°14′19.65″ N, 30 June 2011, *Iberite et Cacciarini*
*Limonium amynclaeum* Pignatti	GA	Lazio region, Gaeta, Trecento scalini, 13°31′50.67″ E, 41°13′9.79″ N, 12 August 2015, *Iamonico*
*Limonium amynclaeum* Pignatti	GI	Lazio region, Gianola, 13°40′29.95″ E, 41°14′47.23″ N, 12 August 2015, *Iamonico*
*Limonium amynclaeum* Pignatti	TE	Lazio region, Terracina, 13°15′47.30″ E, 41°17′20.17″ N, 13 July 2015, *Iberite et Nicolella*
*Limonium circaei* Pignatti	TP	Lazio region, Circeo, Torre Paola *, 13° 2′4.50″ E, 41°14′47.06″ N, 13 July 2015, *Iberite et Nicolella*
*Limonium pontium* Pignatti	IF	Lazio region, Ponza, Il Faro, 12°57′15.90″ E, 40°52′48.09″ N, 29 July 2014, *Iberite et Nicolella*
*Limonium pontium* Pignatti	SS	Lazio region, Palmarola, San Silverio, 12°51′18.56″ E, 40°56′23.08″ N, 27 June 2016, *Iberite et Nicolella*
*Limonium pontium* Pignatti	ST	Lazio region, Santo Stefano, 13°27′4.73″ E, 40°47′29.12″ N, 26 July 2015, *Iberite et Nicolella*
*Limonium pontium* Pignatti	ZA	Lazio region, Zannone, 13°3′38.31″ E, 40°58′17.75″ N, 25 July 2016, *Iberite et Nicolella*
*Limonium pandatariae* Pignatti	AR	Lazio region, Ventotene, Arco, 13°27′4.73″ E, 40°47′29.12″ N, 27 July 2015, *Iberite et Nicolella*

**Table 2 plants-11-03163-t002:** Single genotypes detected with the nuclear markers (ITS1, 5.8S, ITS2) in the *Limonium* populations and type specimens. See Figure 3 for their distribution map.

ITS1 *		ITS2 *										GenBank	Genotype
80	208	384	390	396	420	435	446	449	513	551	583		
TCCCA	-	C	C	C	C	T	C	G	T	C	-	OP452889	**N1**
GGAT-	-	C	C	C	C	C	C	G	T	C	GA	OP452890	**N2**
TCCCA	-	C	C	C	C	T	C	A	T	C	-	OP452891	**N3**
TCCCT	TA	T	T	T	T	T	A	G	C	T	-	OP452892	**N4**

***** The base position corresponds to the genotype N4 sequence as reference. -, deletion.

**Table 3 plants-11-03163-t003:** Haplotypes detected with the plastid markers (*pet*B-*pet*D IGS + petD intron and *trn*L^(UAA)^ intron + *trn*L^(UAA)^-*trn*F^(GAA)^ IGS) in the *Limonium* populations and type specimens. See Figure 4 for the distribution map.

petB-petD *			GenBank	*trn*L^(UAA)^-*trn*F^(GAA)^ *	GenBank	Haplotype
*pet*B-*pet*D IGS		*pet*D intron		*trn*L^(UAA)^ intron		
126	302	742		87		
T	T	C	OP485326	A_6_	OP485330	**A**
T	C	C	OP485327	A_6_	OP485331	**B**
C	T	C	OP485328	A_6_	OP485332	**G**
T	T	T	OP485329	A_5_	OP48533	**R**

* The base position corresponds to the haplotype A sequence as reference.

**Table 4 plants-11-03163-t004:** Morphological characters measured. Qualitative characters are marked with an asterisk (*); the other characters are quantitative (units in millimeters and, for average angle, in degrees).

Height
Number of fertile branches *
Number of sterile branches *
Average angle between branches
Length of the leaf
Width of the leaf
Length of ludicule
Maximum distance among the spikelets
Length of the spike
Number of spikelets *
Length of the spikelets
Length of the outer bract
Width of the outer bract
Length of the median bract
Width of the median bract
Length of the inner bract
Width of the inner bract
Number of flowers per spikelet
Length of the tube
Length of the limb
Length of the corolla lobes
Width of the corolla lobes

**Table 5 plants-11-03163-t005:** MANOVA applied on taxa groups.

Test Statistic	Test Value	F-Ratio	*p* (0.05)
Wilks’ Lambda	0.142926	16.38	0.00000001
Hotelling–Lawley Trace	3.2985	16.35	0.00000001
Pillai’s Trace	1.242707	16.41	0.00000001
Roy’s Largest Root	1.797216	17.97	0.00000001

## Data Availability

The datasets generated in the current study are available from the corresponding author on reasonable request.

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
