# Peer review of "Taxonomy Complexity of Some Tyrrhenian Endemic *Limonium* Species Belonging to *L. multiforme* Group (Plumbaginaceae): New Insights from Molecular and Morphometric Analyses"

_plants, 2022, doi:10.3390/plants11223163_

Round 1

Reviewer 1 Report

The presented manuscript is devoted to phylogenetic, phylogeographic and morphometric study of a group of local endemic species from Lazio region (central Italy) belonging to Limonium multiforme aggr. The authors studied nrITS and four non-coding cpDNA regions. They revealed a fairly low molecular genetic variability, which, however, was slightly geographically structured. The results of morphological analyses demonstrated the presence of three morphological groups, which are not completely genetically isolated from each other. New taxonomic concept of the studied group is presented. All studied taxa of Limonium are accepted as members of one species, Limonium pontium. Within this species, three subspecies have been recognized that do not correspond to previously recognized taxa. The article includes taxonomic treatment of L. pontium and a diagnostic key for a group of studied and related Limonium species. The study is well designed and conducted. The conclusions are confirmed by the results.

I have only minor items to comment:

1.      Please, indicate the specimens and taxa illustrated in the photos (Figure 2).

2.      Page 10, line 339: It seems that these three groups are not separated and form common cloud of dots, as can be seen in Fig. 5. Please describe it more adequately.

3.      Page 11, line 356: Groups are poorly isolated. Please describe it more adequately.

4.      Page 11, lines 370-372: 56% is a pretty low percentage, isn't it?

5.      In the Table 1 you use Population codes (AR, CV, TE, etc.) and in Figure 6 you use locality names. It is quite difficult to compare one with the other. Please, use the same names.

6.      Page 14, Figure 9: It is not explained which parameters are shown as central line, box and whiskers.

 Several comments are also added to the pdf-file.

Author Response

REVIEWER1: Comments and Suggestions for Authors

  1. Please, indicate the specimens and taxa illustrated in the photos (Figure 2).

The name of the taxon was added. However, the specimen cannot be indicated since the pictures were made directly on field without collectiong that plant.

  1. Page 10, line 339: It seems that these three groups are not separated and form common cloud of dots, as can be seen in Fig. 5. Please describe it more adequately.

A new figure 5 was prepared using coloured convex polygons.

  1. Page 11, line 356: Groups are poorly isolated. Please describe it more adequately.

We added “partially overlapped”.

  1. Page 11, lines 370-372: 56% is a pretty low percentage, isn't it?

Yes, is quite low. We sligthly changed the sentence.

  1. In the Table 1 you use Population codes (AR, CV, TE, etc.) and in Figure 6 you use locality names. It is quite difficult to compare one with the other. Please, use the same names.

Populations codes were added to the Figure 6.

  1. Page 14, Figure 9: It is not explained which parameters are shown as central line, box and whiskers.

Explanations was reported in the caption of the Figure 9.

Several comments are also added to the pdf-file.

Replies to the further comments reported in the text follow:

Rev1 says (row 18) “by means of monecular”: “of” was added.

Rev1 says (row 35) “[5])”: “)” was added.

Rev1 says (row 61) “11 Limonium taxa?”: Limonium” was added.

Rev1 says (row 66) “aggregates were created to groups similar microspecies”: “to groups” was corrected as “to group”.

Rev1 says (rows 99-100) “missing cling paranthesis”: “)” was added at the end.

Reviewer 2 Report

In the 'Taxonomic treatment' it would be more obvious if the subspecific names were also in bold type; subsp. pontiumsubsp. terracinensesubsp. pandatariae.  

Even if '515' refers to the publication of Limonium pandatariae, it would be better to cite the place of publication with the 'comb. et stat. nov.'  It's not clear to me what the 'B-' refers to. 

The basionym and place of publication for Limonium pontium Pignatti, Bot. J. Linn. Soc. 64: 264. 1971 subsp. pandatariae.515 (Pignatti) Iamonico, Iberite, De Castro & Nicolella, comb. et stat. nov. B-. Basionym Limonium pandatariae Pignatti, Webbia 36(1): 54 1982. Holotype: Italy, 516 Lazio region, Ventotene island, Montagnozzo, 22.09.1901, A. Béguinot s.n. (RO!)

With the difficulties in the taxonomy of Limonium, can the identifications for the sequences in GenBank be trusted as accurate?  I tend to question GenBank identifications.  At the least, are digital images of the voucher specmens available online?  

Author Response

Reviewer2: Comments and Suggestions for Authors

In the 'Taxonomic treatment' it would be more obvious if the subspecific names were also in bold type; subsp. pontium, subsp. terracinense, subsp. pandatariae.

We agree.

Even if '515' refers to the publication of Limonium pandatariae, it would be better to cite the place of publication with the 'comb. et stat. nov.'  It's not clear to me what the 'B-' refers to.

We do not better understand what the reviewer asks. The format to propose the nomenclatural change is that usually occurring in literature, i.e. binomial+reference+ “comb. et stat. nov.”.

The basionym and place of publication for Limonium pontium Pignatti, Bot. J. Linn. Soc. 64: 264. 1971 subsp. pandatariae.515 (Pignatti) Iamonico, Iberite, De Castro & Nicolella, comb. et stat. nov. B-. Basionym Limonium pandatariae Pignatti, Webbia 36(1): 54 1982. Holotype: Italy, 516 Lazio region, Ventotene island, Montagnozzo, 22.09.1901, A. Béguinot s.n. (RO!)

Many thanks. We added the basionym and its reference.

With the difficulties in the taxonomy of Limonium, can the identifications for the sequences in GenBank be trusted as accurate? I tend to question GenBank identifications. At the least, are digital images of the voucher specmens available online?

We are working to activate the procedure in providing online available specimens deposited at RO.